# Gut Microbiota and Immune System Dynamics in Parkinson’s and Alzheimer’s Diseases

**DOI:** 10.3390/ijms252212164

**Published:** 2024-11-13

**Authors:** Natasa Kustrimovic, Sahar Balkhi, Giorgia Bilato, Lorenzo Mortara

**Affiliations:** 1Department of Biotechnology and Life Sciences, University of Insubria, 21100 Varese, Italy; natasa.kustrimovic1@uninsubria.it; 2Immunology and General Pathology Laboratory, Department of Biotechnology and Life Sciences, University of Insubria, 21100 Varese, Italy; sahar.balkhi@uninsubria.it (S.B.); gbilato@uninsubria.it (G.B.); 3Unit of Molecular Pathology, Biochemistry and Immunology, IRCCS MultiMedica, 20138 Milan, Italy

**Keywords:** microbiota, neurodegenerative diseases, Parkinson’s disease, Alzheimer’s disease

## Abstract

The gut microbiota, a diverse collection of microorganisms in the gastrointestinal tract, plays a critical role in regulating metabolic, immune, and cognitive functions. Disruptions in the composition of these microbial communities, termed dysbiosis, have been linked to various neurodegenerative diseases (NDs), such as Parkinson’s disease (PD) and Alzheimer’s disease (AD). One of the key pathological features of NDs is neuroinflammation, which involves the activation of microglia and peripheral immune cells. The gut microbiota modulates immune responses through the production of metabolites and interactions with immune cells, influencing the inflammatory processes within the central nervous system. This review explores the impact of gut dysbiosis on neuroinflammation, focusing on the roles of microglia, immune cells, and potential therapeutic strategies targeting the gut microbiota to alleviate neuroinflammatory processes in NDs.

## 1. Introduction

The human microbiota is a complex community of bacteria, bacteriophages, viruses, fungi, and protozoa that varies significantly between individuals and across different body sites, such as the skin, oral cavity, genitourinary tract, and gastrointestinal tract [1]. It plays a crucial role in various physiological functions, including inflammation, metabolism, hematopoiesis, and cognitive functions. Dysbiosis, an imbalance in microbial populations, can result from factors like antibiotic use, dietary changes, infections, chronic stress, environmental pollutants, chronic diseases, aging, and lack of physical activity, leading to potential health issues, including inflammatory diseases and cancer [2].

The gut microbiota is established early in life, with colonization beginning at birth, increasing in diversity during the first five years, and stabilizing with age [3]. Studies using 16S rRNA sequencing and metagenomics have identified the dominant gut microbial phyla: *Firmicutes* and *Bacteroidetes*, followed by *Actinobacteria*, *Proteobacteria*, *Fusobacteria*, and *Verrucomicrobia* [4,5,6]. Within these phyla, *Clostridium* dominates the *Firmicutes*, while *Bacteroides* and *Prevotella* are prevalent in the *Bacteroidetes*.

Despite the interindividual differences in species composition, the gut microbiota in healthy individuals exerts a relatively consistent array of metabolic functions that are referred to as the “core microbiota” [7]. In addition to their central role in regulating the digestive process (extraction, synthesis, and absorption of numerous nutrients and metabolites), commensal bacteria within the gut microbiota, together with their products, have a critical role in regulating the development, homeostasis, and function of innate and adaptive immune cells [8]. Furthermore, gut bacteria play a pivotal role in immune modulation and the development of the nervous system [9,10,11].

Dysbiosis of the gut microbiota has been implicated in numerous disorders, ranging from intestinal diseases, such as colorectal cancer and inflammatory bowel disease (IBD), to more systemic diseases, such as diabetes, metabolic syndrome, atherosclerosis, hypertension, and cystic fibrosis [12]. Furthermore, a substantial amount of research has also linked microbial dysbiosis to neurodegenerative disorders, such as Parkinson’s disease (PD), Alzheimer’s disease (AD), and multiple sclerosis (MS) [12].

## 2. The Microbiota–Gut–Brain Axis

The microbiota–gut–brain (MGB) axis, an intricate bidirectional communication pathway that links the brain and the gastrointestinal system, encompasses diverse mechanisms of communication. The rapid progress in microbiota science has revolutionized the understanding of the MGB axis, unveiling the interactions involved in this axis, including the gut-associated immune system, enteric neuroendocrine system (ENS), and the gut microbiota. The brain, the gut, and microbiota are the three nodes in the MGB network, and the communication between them is bidirectional through multiple pathways: neural, endocrine, metabolic, and immune [13]. The communication between three nodes of the MGB axis is established through the autonomic nervous system (ANS), the enteric nervous system (ENS), the hypothalamic–pituitary–adrenal axis (HPA), and the gut microbes [14]. Disruptions that may occur in the MGB axis have been implicated in the pathophysiology of neurodegenerative disorders [15] (Figure 1).

Interoception, the brain’s ability to sense internal physiological states, is crucial for homeostasis and linked to various disorders. It influences emotional states and homeostatic reflexes, maintaining balance within the MGB axis [16]. Gut microbial signals can modulate these reflexes by acting on ENS neurons or via systemic circulation, with the CNS responding through the ANS and HPA axes [17,18]. The ANS affects gastrointestinal physiology, including barrier integrity, motility, secretion, and immune response, impacting gut microbiota diversity [19]. Neurotransmitters like norepinephrine and dopamine from sympathetic neurons can affect microbial gene expression and growth [20]. Gut microbiota also influences brain function by modulating neurotransmitter synthesis, such as acetylcholine, serotonin, norepinephrine, dopamine, and glutamate [21,22,23]. The ENS, often called the “second brain”, autonomously governs the gastrointestinal tract (GIT) functions and communicates bidirectionally with the CNS via the sympathetic and parasympathetic systems. It transmits gut microbiota changes to the brain through neural pathways, including the vagus nerve [24,25,26]. This communication involves neuropeptides, neurotransmitters, cytokines, and microbial metabolites [27].

The gut also communicates with the HPA axis. In response to environmental stimuli, the hypothalamus releases corticotrophin-releasing hormone (CRH), triggering adrenocorticotrophic hormone secretion and cortisol release, which disrupts intestinal permeability and alters the gut microbiota [28,29].

Gut–brain communication also involves hormonal signaling through gut peptides released by enteroendocrine cells (EECs). EECs detect various gut luminal contents and interact with the microbiota to secrete hormones affecting brain function. These hormones, such as orexin, galanin, ghrelin, gastrin, and leptin, influence brain regions like the area postrema, impacting feeding behavior, energy balance, sleep–wake cycle, sexual behavior, arousal, and anxiety [30,31].

Overall, the intricate relationships within the microbiota–gut–brain axis are foundational to both physiological and psychological health. The bidirectional pathways involving the ENS, ANS, and HPA axes demonstrate how the gut microbiota directly influences brain function and behavior through diverse mechanisms, including hormonal and neural communication. Disruptions in this axis may contribute to the pathophysiology of both gastrointestinal and NDs.

## 3. Gut Microbiota and Immune System

The colonization of the intestinal tract by diverse microbes profoundly impacts both the innate and adaptive immune systems [32]. Early studies on germ-free (GF) animals demonstrated that the absence of commensal microbes leads to significant defects in intestinal lymphoid tissue architecture and immune function, such as reduced αβ and γδ intra-epithelial lymphocytes [33] and the absence of inducible Th17 cells [34]. The gut microbiota significantly influences adaptive immune responses by affecting T and B cell development and distribution. For instance, a polysaccharide derived from the commensal *Bacteroides fragilis* directs the maturation of the immune system, correcting systemic T cell deficiencies and Th1/Th2 imbalances [35]. Additionally, extracellular signals from commensal microbes regulate an early B cell lineage in the intestinal mucosa, influencing gut immunoglobulin repertoires [36]. Microbiome-derived ligands, such as toll-like receptors (TLRs) and nucleotide-binding oligomerization domain (NOD)-like receptors (NLRs), along with metabolites like short-chain fatty acids (SCFAs) and AhR ligands, directly affect enterocytes and intestinal immune cells while also reaching distant tissues to influence systemic immunity. In Peyer’s patches, Foxp3^+^ regulatory T cells (Tregs), immunoglobulin A, and Th17 cells facilitate B cell class switching and the production of secretory IgA, which helps regulate and compartmentalize gut microbiota [37] and promotes the expansion of Foxp3^+^ Tregs [38]. The microbiota also plays a crucial role in regulating CD8^+^ (cytotoxic) T cell responses, which are essential for eliminating intracellular pathogens and cancer cells. CD8^+^ T cells require professional antigen-presenting cells (APCs) for priming and rely on CD4^+^ T cell signaling for amplification. However, in GF mice, antigen-activated CD8^+^ T cells do not transition into memory cells; this process necessitates microbiota-derived SCFAs, underscoring the microbiome’s importance in shaping long-term CD8^+^ T cell responses [39]. Invariant natural killer T cells (iNKTs) are another immune cell group influenced by the gut microbiota. In GF mice, iNKTs exhibit a less mature phenotype and reduced activation in response to antigens, indicating that microbial colonization is critical for their proper maturation and function [40]. Recent research has begun to elucidate the interactions between the microbiota and tissue-resident dendritic cells (DCs). A newly identified Syk kinase-coupled signaling pathway in DCs is crucial for the microbiota-induced production of IL-17 and IL-22 by CD4^+^ T cells, emphasizing the role of DCs in linking microbial signals to adaptive immune responses [41]. The intestinal microbiota induces TH17 cytokine production by interacting with DCs. Specifically, certain gut bacteria, like segmented filamentous bacteria (SFB), activate DCs in Peyer’s patches (PPs) through pattern recognition receptors such as Mincle. This triggers the secretion of key cytokines like IL-6 and IL-23 by DCs, which are essential for the differentiation of naïve T cells into TH17 cells. TH17 cells, in turn, produce IL-17 and IL-22, maintaining intestinal barrier integrity by regulating immune responses and preventing microbial translocation [41]. Moreover, monocytes and macrophages, vital innate immune cells, also interact with the gut microbiota. Additionally, a SCFA metabolite produced by the microbiota, butyrate, promotes the differentiation of monocytes into macrophages by inhibiting histone deacetylase 3 (HDAC3), thereby enhancing the host’s antimicrobial defense capabilities [42]. Furthermore, various immune cells present in the gut, such as Paneth cells, DCs, and macrophages, interact with gut microbiota through receptors like TLRs and NLRs. These receptors detect microbial patterns and regulate immune responses, which are crucial for maintaining a balanced, non-inflammatory state. Mechanisms like mucosal barriers, antimicrobial proteins, and secretion of IgA also contribute to immune tolerance [43,44].

## 4. Neurodegenerative Disorders

NDs involve the progressive loss of neuronal function in the brain and spinal cord, leading to cognitive and motor decline [45]. Common NDs include Parkinson’s disease (PD), Alzheimer’s disease (AD), amyotrophic lateral sclerosis (ALS), multiple sclerosis (MS), frontotemporal dementia (FTD), and Huntington’s disease (HD). These disorders result from genetic, environmental, and lifestyle factors, with aging as a significant risk factor [46,47]. As life expectancy increases, so does ND prevalence, imposing greater societal burdens [48]. Despite varied etiologies, many NDs share common pathological features such as abnormal protein aggregation, mitochondrial dysfunction, oxidative stress, and neuroinflammation [49].

Almost all NDs are characterized by the accumulation of intra- or extracellular proteins in the CNS [50]. In physiological conditions, those proteins exist in unstructured forms, but in the context of NDs, these proteins undergo significant conformational changes, leading to alterations in their structural folding and the formation of both oligomeric and fibrillary aggregates [51]. These modifications in size and three-dimensional shape facilitate self-association and precipitation in specific brain regions, resulting in the acquisition of pathological protein characteristics. Misfolded protein conformational changes can occur due to post-translational modifications, impaired protein clearance, or increased protein production [51]. It has been reported that extracellular and intracellular protein aggregates and misfolded proteins may function as pathogen-associated molecular patterns (PAMPs), leading to chronic activation of the innate immune response via pattern recognition receptors (PRRs) [52,53]. Protein deposits are capable of activating a variety of PRRs, including TLRs, formyl peptide receptors, receptors for advanced glycation end products, scavenger receptors, complement, and pentraxins [54], resulting in the acute neuroinflammation mainly characterized by activation of microglia in CNS and its polarization towards the pro-inflammatory M1 phenotype, characterized by extensive production of a variety of pro-inflammatory mediators. If the clearance of the misfolded proteins or aggregates is not decisive, the acute neuroinflammation tends to become chronic with prolonged microglial activation, excessive pro-inflammatory cytokine production, and increased oxidative stress, all further leading to the activation of the adaptive immune response and subsequently resulting in neuronal death [55,56,57,58,59,60,61].

In neuroinflammation, a pivotal role has been ascribed to the activated microglial cells. The pro-inflammatory M1 subtype of microglial cells, induced by TLRs and gamma interferon (IFNγ) signaling, releases pro-inflammatory cytokines like IL-1β, IL-6, TNF-α, and NF-kappa B and expresses NADPH oxidases and matrix metalloproteinase-12 (MMP-12) [62], and has a significant role in the neuroinflammation of astrocytes as well. When neuroinflammation occurs, astrocytes undergo a reactive transformation marked by morphological alterations and increased production of glial fibrillary acidic protein. In this reactive state, astrocytes can generate and release various inflammatory agents like cytokines and chemokines, which impact the activities of neighboring neurons and immune cells. This response can be beneficial, aiding in pathogen and waste removal while supporting repair mechanisms [63], or it can be detrimental, aiding in the perpetuation of the inflammation and prompting the initiation of the NDs. Furthermore, in NDs, dysfunctional astrocytes can impede neuronal metabolic support, exacerbate excitotoxicity by reducing glutamate uptake, and disrupt synaptic activity, all contributing to neuronal degeneration and cognitive decline [64]. Nevertheless, the continuation of the neuroinflammatory process in the CNS is significantly aided by infiltrated peripheral immune system cells.

The CNS, once considered immune-privileged, can be infiltrated by immune cells during peripheral inflammation, such as exposure to lipopolysaccharides (LPS) or viral infections [65]. Neuroinflammation starts with microglial activation, releasing pro-inflammatory messengers, weakening the BBB, and allowing peripheral immune cell entry [66] (Figure 2).

The BBB is a key regulator of CNS homeostasis and is highly related to the function of microvascular endothelial cells together with microglia, astrocytes, neurons, and constituents of the extracellular matrix. This cellular component network is known as the neurovascular unit (NVU), and activated microglial cells seem to be the main regulators for dynamic remodeling of the BBB. In NDs, M1 pro-inflammatory microglia contribute to BBB dysfunction and vascular “leak”, while M2 anti-inflammatory microglia play a protective role. Peripheral immune cells, including monocytes, neutrophils, NK cells, DCs, T cells, and B cells, each serve unique roles in the immune response within the CNS. Monocytes are highly plastic cells that can adapt to diverse microenvironments and are involved in both protective and pathogenic responses in the CNS. Upon injury or inflammatory stimuli, monocytes are recruited to the CNS, where they can differentiate into macrophages and contribute to the immune response alongside resident microglia. Their migration into the brain is regulated by the CCL2–CCR2 axis, which facilitates their crossing of the BBB in neuroinflammatory conditions [67]. In AD, monocytes infiltrate the brain and phagocytize amyloid-beta (Aβ) deposits, potentially reducing the toxic burden [68]. In PD, upregulation of CCR2 in peripheral monocytes has been linked to their migration into the inflamed brain [69]. Neutrophils can also migrate into the CNS under pathological conditions such as neurodegeneration, releasing substances that break down the BBB and exacerbating its permeability, attracting further neutrophil infiltration that, by secreting pro-inflammatory cytokines such as IL-17, intensify the damage [70]. NK cells, guided by chemokines produced during neuroinflammation, migrate to the CNS and can eliminate glial cells, affecting CNS function [71]. DCs migrate to areas like the meninges or choroid plexus to present antigens to T cells, triggering an immune response. Furthermore, B cells increase in number in the CNS during inflammation, playing a significant role in the immune processes within the inflamed CNS [70]. Importantly, these peripheral immune cells share important functional characteristics with microglia, such as the expression of TLRs, which is important since this enables their re-activation in the CNS by aggregated proteins. Reactivated immune cells release significant quantities of pro-inflammatory cytokines, thus exacerbating the BBB breakdown and further activation of resident microglial cells, further contributing to the perpetuation of the neuroinflammatory process [70].

The role of the gut microbiota in NDs is still not well understood. Given the significant role of neuroinflammation and the immune system in NDs, the gut microbiota’s involvement can be understood through its impact on immune system development and activation. The gut microbiota communicates with the brain via cytokines, chemokines, and microbial-associated molecular patterns (MAMPs). For instance, bacterial peptidoglycan and LPS from Gram-negative bacteria are crucial in this communication. The immune system recognizes bacterial LPS via TLR4, which is present in microglia, the brain’s primary immune cells. LPS can travel through the bloodstream to the brain under pathological conditions, triggering neuroinflammatory responses [72]. Additionally, the gut microbiota influences microglial maturation and function through bacterial-derived SCFAs like acetate, butyrate, and propionate. Germ-free mice, lacking typical gut microbiota, show increased numbers of immature microglia across various brain regions [73]. The impact of microbial elements on microglial development and function varies by host sex, highlighting sex-specific differences in susceptibility to certain CNS disorders [74].

## 5. Parkinson’s Disease

Parkinson’s disease (PD) is a progressive neurodegenerative movement disorder affecting 1–2% of individuals over 65 years old, making it the second most common neurodegenerative disorder after Alzheimer’s disease [75]. PD is characterized by the degeneration of melanin-containing, dopaminergic neurons in the substantia nigra pars compacta (SNpc), which project to the corpus striatum, crucial for regulating posture and muscle tone [76]. Aside from the dopaminergic system, PD also involves other neurotransmitter systems, such as the noradrenergic, serotonergic, and cholinergic systems [77]. A central pathological feature of PD is the presence of α-synuclein (α-syn) lesions, which form aggregated fibrils with abnormal tertiary structures known as Lewy bodies (LB). The misfolding and aggregation of α-syn, encoded by the SNCA gene, are thought to drive progressive neurodegeneration in PD [78] (Figure 3). Motor symptoms of PD include resting tremor, bradykinesia, rigidity, and postural instability [79,80], while non-motor symptoms such as autonomic nervous system dysfunction lead to gastrointestinal issues, bladder dysfunction, sialorrhea, excessive sweating, and orthostatic hypotension [81]. Depression, cognitive decline, impaired visual-spatial perception, attention deficits, and dementia are also common [80]. Determining the etiology of PD is challenging as clinical symptoms manifest after significant dopaminergic neuron loss, typically exceeding 70% [82]. Age is a major risk factor, with PD prevalence increasing to 1–2% at age 65 and 3–5% by age 85 [83]. Genetic predisposition, environmental toxins, head trauma, and infections also contribute to PD risk [84]. Current PD treatments are predominantly symptomatic, focusing on dopamine replacement therapies like L-DOPA and dopaminergic agonists [85,86,87]. Non-dopaminergic therapies are also used or under evaluation to address motor symptoms [87]. Efforts continue to develop treatments targeting the disease mechanisms to potentially prevent PD, slow its progression, and promote neuroprotection.

### 5.1. Gut Microbiota and α-Synuclein and Immune System Interaction in PD

The gut microbiota plays an important role in the pathogenesis and progression of PD through multiple mechanisms.

The “dual-hit hypothesis” proposed by Heiko Braak suggests that PD pathology may initiate in the gastrointestinal tract due to exposure to toxins or microbial pathogens. This triggers the pathological formation of α-syn in nerve cells of the submucosal plexus. According to this theory, α-syn aggregates begin in the ENS or olfactory bulb before spreading to the substantia nigra and other CNS regions [88,89]. Animal studies support this hypothesis, demonstrating retrograde transport of α-syn via the vagus nerve to brainstem nuclei and the substantia nigra, resulting in dopaminergic neuron loss [90,91,92]. Exosomes and nanotubules are considered potential mechanisms for this transport [93]. Notably, vagotomy has been shown to prevent the retrograde propagation of α-syn from the peripheral ENS to the CNS in some studies [90,91,92]. Additionally, a study examining the impact of complete vagotomy on the risk of developing PD showed a significant reduction, though not complete elimination, of the risk. This suggests the existence of alternative transmission routes, such as the olfactory pathway [94]. Intriguingly, the presence of α-syn in the colon, duodenum, and stomach of PD patients has been detected 5–8 years before the onset of initial motor symptoms [95,96], indicating that accumulated α-syn in the bowel could serve as a biomarker for early PD detection.

Hasuike et al. (2019) stated that α-syn in PD patients can be found in the gastrointestinal tract nerve plexuses and that these depositions increased intestinal permeability and levels of Gram-negative bacteria such as *E. coli* [97], which is in accordance with previously published studies that have recorded frequently present gut inflammation and increased intestinal permeability in PD patients [98,99]. This inflammatory state is marked by increased levels of LPS-binding protein in the plasma, increased pro-inflammatory cytokine production in both the colon and glial cells, as well as activation and structural changes in the epithelial barrier, such as the downregulation of specific tight junction proteins [100,101]. Furthermore, in vitro tests have shown that both monomeric or fibrillar forms of α-syn exhibit chemotactic activity towards isolated human neutrophil and monocyte cultures. These forms can also promote dendritic cell maturation [102] and influence the distribution of peripheral CD4^+^ T cells, T central memory cells, and T effector memory cells in PD patients [103]. Interestingly, animal studies have shown that intraperitoneal administration of low-dose LPS induces α-syn deposition in the colons of treated rats, along with increased intestinal permeability [104]. Hence, it can be concluded that the accumulation of α-syn inclusions in the gut may directly contribute to local inflammation, exacerbating gut dysfunction in clinical and preclinical PD.

Recently, it has been reported that *E. coli* produces an amyloid protein called “curli”, which, when hybridized with human amyloid in vitro, exacerbates α-syn pathology in mice [105], leading the authors to hypothesize that alterations in gut microbial balance may damage the gut barrier, prompting a protective immune response in humans while also increasing pathological α-syn expression in the gut nervous system. Repeated oral administration of curli-producing bacteria to aged rats induced increased neuronal α-syn deposition in both the gut and the brain tissues [106]. In addition to *E. coli* and *Pseudomonas*, numerous other bacteria that reside in the gut, such as *Streptoccocus*, *Staphylococcus*, *Salmonella*, *Mycobacteria*, *Klebsiella*, *Citrobacter*, and *Bacillus* species, are capable of making extracellular amyloids. The pathogenic role of amyloid produced by the gut microbiota in the development of PD has been demonstrated through studies of fecal microbiota transplantation from PD patients into mice in which pathological α-syn aggregation and neuroinflammation were promoted, and PD motor symptoms were aggravated [107,108].

In addition to the potential for direct pathogenic cross-seeding between amyloids from different organisms, bacterial amyloids are recognized as microbe-associated molecular patterns (MAMPs), capable of directly stimulating and priming the host’s immune response. This activation can contribute to pro-inflammatory conditions in the gut, creating an environment conducive to protein aggregation, cellular dysfunction, and cell death. Inflammatory signals originating from the gut can reach the brain through immune cell infiltration or indirectly, suggesting that human α-synuclein (α-syn) could be mistaken for a microbe-associated molecular pattern (MAMP), mimicking bacterial amyloids and amplifying inflammation [109]. This process is supported by findings in the brain, where oligomeric α-syn binds to Toll-like receptor 2 (TLR2) on microglia, leading to the production of pro-inflammatory cytokines such as TNF and IL-1β [110,111]. Moreover, TLR4 expression on microglia is required for the phagocytosis of α-syn [112]. This brain-based immune response mirrors what occurs in the gut, where amyloid proteins like α-syn trigger a similar activation of immune pathways. In the gut, the presence of α-syn and other amyloids is associated with robust immune activation. Specifically, the TLR2/1/CD14 heterocomplex recognizes curli proteins, activating the NF-κB pathway and inducing the production of pro-inflammatory cytokines such as IL-8, IL-6, IL-17A, and IL-22 [113,114,115,116]. Curli proteins also activate the NLRP3 inflammasome, leading to caspase-1/11 activation and the maturation of pro-IL-1β into IL-1β [117]. This shared inflammatory response between the gut and brain highlights how local gut inflammation may contribute to α-syn aggregation, creating a feedback loop that drives both gut and brain pathology.

Dysbiosis of the gut microbiota disrupts the BBB by altering tight junction proteins, facilitating the infiltration of harmful substances into the CNS [118]. Chronic gut inflammation and increased gut permeability lead to the increased release of pro-inflammatory cytokines, such as IL-1β, IL-6, and TNF-α, that can cross the BBB via the gut–brain axis, inducing neuroinflammation and neuronal death [119]. Alterations in microbiota composition also affect the levels of metabolic products, neuroactive substances, and bioactive substances. SCFAs, produced by the gut microbiota, serve as an energy source and exert neuroprotective effects by upregulating glial cell line-derived neurotrophic factor (GDNF) and brain-derived neurotrophic factor (BDNF) [120]. Gut dysbiosis may also influence PD by altering ghrelin levels. Changes in gut microbiota have been associated with variations in plasma ghrelin levels, which are typically lower in PD patients [121,122]. Ghrelin, known for its appetite-stimulating effects, also exhibits neuroprotective properties in PD and AD through the activation of adenosine monophosphate kinase [123].

### 5.2. Changes in Gut Microbiota in PD

The progression of PD has been linked to gut microbiota dysbiosis. Studies have shown an increased abundance of *Lactobacillus* and *Bifidobacterium* genera in PD [124] along with decreased levels of butyrate-producing bacteria such as *Faecalibacterium*, *Coprococcus*, *Blautia*, and *Roseburia*, which possess anti-inflammatory properties [125,126,127,128,129]. Cirstea et al. (2020) reported an increase in *Akkermansia* and *Bifidobacterium* and a decrease in *Lachnospiraceae* and *Faecalibacterium* in fecal samples from PD patients compared to healthy controls [130]. Aho et al. (2019) investigated gut microbiota changes in PD patients, identifying significant differences, including decreased *Prevotella* in faster-progressing patients [131]. Pietrucci et al. [132] used machine learning to classify microbiota data from PD patients, confirming increased *Enterobacteriaceae* abundance and decreased *Lachnospiraceae*, previously identified using 16s RNA sequencing [133]. These alterations were correlated with PD severity and motor impairment. Petrov et al. (2017) used 16S rRNA to analyze the bacterial genomes in PD patients and found lower alpha diversity levels compared to the controls [128]. They attributed these findings to latent inflammation in the intestine, which is known to trigger misfolding of α-syn in gut neurons. PD patients showed increased levels of *Catabacter*, *Lactobacillus*, *Christensenella*, *Oscillopsia*, and *Bifidobacterium* (Table 1). The study highlighted a decrease in taxonomic diversity, detecting only nine genera and 15 species of microorganisms in PD patients [128].

Several studies, including those by Keshavarzian et al. [129] in the USA, Li et al. [134] in China, Unger et al. [135] in Germany, and Hasegawa et al. [136] in Japan, reported a mild reduction in *Prevotella* abundance among PD patients. In contrast, studies by Scheperjans et al. [126] in Finland, Bedarf et al. [137] in Germany, and Petrov et al. [128] in Russia noted a significant decrease in *Prevotella* species in PD patients. Interestingly, Hill-Burns et al. [138] in the USA and Barichella et al. [139] in Italy did not find an association between *Prevotella* and PD.

The reduced abundance of *Prevotellaceae* compared to the elevated levels of *Enterobacteriaceae* in the stool of PD patients holds significant implications. *Prevotellaceae* normally act as beneficial commensal bacteria, aiding in mucin production and generating SCFAs through dietary fiber fermentation [6]. Their decrease can lead to increased gut permeability, exacerbating exposure to bacterial endotoxins and triggering inflammation. This amplified permeability allows bacterial toxins to induce excessive α-syn expression in the colon, promoting its misfolding [140]. Conversely, an increase in *Enterobacteriaceae* elevates levels of LPS, derived from the cell walls of these Gram-negative bacteria, were identified [140]. In conjunction with a decreased abundance of *Prevotellaceae*, this can facilitate the passage of LPS and other neurotoxins across the intestinal barrier into the bloodstream, as evidenced by elevated levels of LPS-binding protein in PD patients’ blood [141,142]. This process further disrupts the intestinal epithelial barrier and triggers inflammation. Furthermore, systemic circulation of LPS can provoke systemic inflammation by activating pathways like TLR4 and NF-κB, leading to the production of pro-inflammatory cytokines such as TNF-α, IL-1β, IL-6, and IL-2, that are capable of disrupting the BBB, contributing to neuroinflammation [142,143,144,145]. Therefore, the relative increase in *Enterobacteriaceae* and decrease in *Prevotellaceae* may directly contribute to the initiation and perpetuation of neuroinflammatory processes within the CNS.

**Table 1 ijms-25-12164-t001:** Bacterial abundance changes in PD patients.

Increased Bacterial Abundance	Decreased Bacterial Abundance
*Enterobacteriaceae* [109,126,133,135,146]	*Lachnospiraceae* [138]
*Akkermansia* [130]	*Faecalibacterium* [127,130,131,134]
*Bifidobacterium* [130,136,147]	*Blautia* [127,128,134]
*Christensenella* [128]	*Dorea* [128]
*Catabacter* [128]	*Bacteroides* [128,135]
*Oscillospiraceae* [128]	*Prevotellaceae* [126,128,131,132,134,135,137]
*Ruminococcus bromii* [128]	*Coprococcus* [127,129]
*Papilibacter cinnamivorans* [128]	*Faecalibacterium* [127,134]
*Proteus* [148]	*Roseburia* [129]
*Lactobacillaceae* [126,128,136,149]	*Ralstonia* [129]
*Clostridium Coccoides* [147]	*B. Fragilis* [147]
*Christinesella* [141]	*Ruminococcus* [134]
	*Oscillospiraceae* [134]

### 5.3. Microbiota–Gut–Brain Axis for Parkinson’s Disease Treatment

Currently, symptomatic treatment for PD primarily aims to improve clinical manifestations and enhance patients’ quality of life, with levodopa being the mainstream therapy. However, patients often develop tolerance over time, necessitating higher doses that can lead to motor function abnormalities. Levodopa primarily addresses the motor symptoms and does not halt disease progression, leaving most non-motor symptoms unaffected. Additionally, severe gut dysfunction in PD patients significantly impairs the absorption of therapeutic drugs.

Recent proposals have focused on treating PD through the microbiota–gut–brain axis. One intervention involves using probiotics or specific bacterial strains beneficial to PD patients. Animal studies have identified several promising strains, including *Bacillus subtilis*, which inhibits α-syn [150], *Clostridium butyricum* [151], and *Lactobacillus* [152].

In 2019, a randomized, double-blind, placebo-controlled clinical trial evaluated the effects of probiotics (*Lactobacillus acidophilus*, *Bifidobacterium bifidum*, *Lactobacillus royi*, and *Lactobacillus fermentum*) in 60 PD patients over 12 weeks. Probiotic supplementation positively impacted motor function, insulin metabolism, and oxidative stress parameters in PD patients [153].

Another potential therapeutic approach is fecal microbiota transplantation (FMT), the most direct way to alter the gut microbial environment. A prospective, single study in PD patients found that FMT restored the overgrowth of gut microbiota, with an increased abundance of *Blautia* and *Prevotella* and a marked decrease in the abundance of *Bacteroidetes,* resulting in a significant decline of scores for the Parkinson’s Disease Rating Scale (UPDRS) and the non-motor symptoms questionnaire (NMSs) [154]. Several clinical trials regarding FMT are on-going (ClinicalTrials.gov Identifier: NCT04837313; Identifier: NCT03808389; Identifier: NCT03876327; Identifier: NCT05204641).

Lastly, the dietary intervention should not be omitted. A Western diet (WD) high in fat and sugar can increase the abundance of microorganisms that produce harmful substances like LPS, which can lead to gut microbiota dysbiosis and increased gut permeability, damaging the BBB and causing neuroinflammation [155]. On the other hand, the Mediterranean diet (MD), rich in vegetables, nuts, and olive oil, can exert neuroprotective effects [156].

## 6. Alzheimer’s Disease

Alzheimer’s disease (AD) is a progressive, chronic, multifactorial neurodegenerative disorder predominantly associated with aging. It is recognized as the most common cause of dementia, significantly impairing cognitive function and daily living activities among the elderly. Globally, AD is a critical public health concern, affecting approximately 13% of individuals aged 75 to 84 and about 33.3% of those aged 85 or older [157,158,159]. The clinical manifestations of AD typically begin with the gradual deterioration of cognitive functions, including memory loss, language difficulties, and impaired reasoning and judgment, accompanied by personality changes. These symptoms progressively worsen, leading to severe cognitive and functional decline [157,158,159,160]. Pathologically, AD is characterized by the accumulation of extracellular amyloid β-peptide (Aβ) and intracellular neurofibrillary tangles (NFTs) composed of hyperphosphorylated tau protein. These pathological features contribute to neuronal damage, oxidative stress, synaptic dysfunction, and neuroinflammation, ultimately leading to synaptic and neuronal loss that ultimately affects memory, reasoning, and other cognitive functions and results in brain atrophy [161,162,163] (Figure 4).

As in PD and AD, neuroinflammation in the CNS plays a significant role, with pro-inflammatory M1 microglia being the most prominent player. The amyloid cascade–inflammation hypothesis suggests that the activation of microglia by Aβ can trigger inflammatory reactions that ultimately lead to the aggregation of tau protein. Tau, primarily found in neurons, plays a role in regulating microtubule assembly and stability. In AD, tau undergoes various post-translational modifications, notably hyperphosphorylation, leading to its disassociation from microtubules, aggregation, and accumulation within neurons, ultimately impairing neuronal function and causing synaptic dysfunction [164].

### 6.1. Interaction Between Gut Microbiota and Immune Cells in AD

Emerging evidence suggests that the gut microbiota may influence brain function and behavior, potentially affecting the pathogenesis of AD.

In the context of AD, pro-inflammatory gut microbiota can exacerbate neurodegenerative processes. For example, *Collinsella*, a bacterial genus, has been identified as a risk factor for AD and is associated with the APOE4 allele, a major genetic risk factor for sporadic AD [165]. Conversely, other genera like *Eubacterium nodatum* and *Prevotella* appear to offer protective effects against AD, suggesting that specific microbial populations may modulate disease progression through their influence on immune and inflammatory pathways [165].

Another notable immune interaction involves bacterial amyloids, such as the amyloid curli produced by *Escherichia coli*. While structurally distinct from neuronal amyloids, bacterial amyloids can activate immune pathways and may prime the immune system to form endogenous amyloids in the brain, contributing to AD pathology. Animal models exposed to curli-producing *E. coli* have shown increased levels of neuronal α-syn and inflammatory markers like TNF-α and IL-6, further linking gut-derived amyloids to neuroinflammation [164,166].

In addition to direct immune modulation, gut bacteria also influence central CNS function through neurotransmitter regulation. Certain bacteria produce or consume GABA, influencing peripheral GABA levels and potentially modulating CNS signaling. Imbalances in GABAergic pathways have been associated with neurodegenerative disorders, including AD [167]. Similarly, gut microbiota regulates serotonin production, with the dysregulation of serotonin metabolism contributing to cognitive decline in AD patients [168,169].

The NLRP3 inflammasome represents another key link between peripheral and central inflammation. Activated by microbial metabolites, the NLRP3 inflammasome contributes to the neuroinflammation observed in AD and other neurodegenerative diseases [170,171]. This microbiota–gut–inflammasome–brain axis underscores the dynamic interaction between gut microbiota and CNS inflammation, offering new insights into potential therapeutic targets for AD. Additionally, studies on laboratory animals, particularly germ-free rodents, have demonstrated the role of gut bacteria in AD pathogenesis, showing a significant decrease in Aβ pathology that is restored upon the reintroduction of gut bacteria [172]. In humans, bacterial or viral infections have been linked to AD. Persistent infection by Helicobacter pylori in AD patients is linked to lower cognitive scores and increased production of inflammatory mediators [173]. AD patients infected with H. pylori, Chlamydia pneumoniae, and Borrelia burgdorferi have higher levels of Aβ in their blood [172,173]. H. pylori filtrate induces tau hyperphosphorylation in neuroblastoma cells, and significant amounts of bacterial LPS have been found in the brains of AD patients [174].

Gut bacteria impact AD through metabolites like SCFAs, which disrupt the protein–protein interactions necessary for Aβ assembly formation [174]. The bacterial metabolite trimethylamine N-oxide (TMAO) increases β-secretase activity, promoting Aβ buildup and worsening AD pathogenesis; it also enhances the transport of Aβ from platelets to the brain and releases calcium ions, leading to platelet hyperreactivity [174]. These findings suggest the potential for developing personalized dietary therapies to manage Aβ development and aggregation in AD. Further evidence of gut microbiota involvement in AD includes studies showing that FMT from healthy mice to AD mice reduces glial responses, amyloid plaques, neurofibrillary tangles, and cognitive impairment [175,176]. FMT, which involves transferring healthy gut bacteria from a donor to a recipient, significantly reduces circulating inflammatory monocytes in AD animals and restores the normal expression of the genes linked to intestinal macrophage activity [177].

### 6.2. Microbiota–Gut–Brain Axis for Alzheimer’s Disease Treatment

The intricate relationship between gut microbiota and AD has sparked significant interest in developing targeted therapies aimed at restoring gut homeostasis. Various strategies, particularly MGB axis-based multi-therapies, offer promising avenues for intervention. These approaches encompass probiotics, prebiotics, synbiotics, and FMT and are complemented by moderate-intensity exercise, a nutritious diet, and sufficient sleep. This multifaceted approach aims to synergistically enhance cognitive function in AD patients [178,179].

Diet plays a crucial role in modulating gut microbiota, influencing both systemic health and neurodegenerative diseases like AD. The Mediterranean diet (MD), rich in fruits, vegetables, legumes, and cereals, is linked to delayed AD progression by 1.5 to 3.5 years [180]. Its beneficial effects are attributed to positive changes in the gut microbiota and anti-inflammatory properties.

The MD promotes the production of SCFAs, such as butyrate, which enhance gut health and reduce inflammation. Increased SCFAs and decreased levels of trimethylamine N-oxide (TMAO), a harmful metabolite linked to neurodegeneration, are notable benefits of this diet. Conversely, diets high in sugar, saturated fats, and processed foods encourage pro-inflammatory gut bacteria, contributing to neuroinflammation and cognitive decline [181].

Polyphenols, found in plant-based foods like fruits and vegetables, further support beneficial microbes such as *Bifidobacterium* and *Lactobacillus*, which are associated with brain health and reduced AD risk. Additionally, plant-based proteins enhance the growth of probiotic bacteria, while animal-derived proteins increase harmful bacteria that may produce neurotoxic by-products [182,183].

Prebiotics, derived from plant fibers, nurture the gut microbiota by serving as substrates for beneficial bacteria like *Lactobacilli* and *Bifidobacteria*, which ferment them to produce SCFAs and other by-products that support beneficial bacteria and inhibit pathogens, promoting gut health [184]. Prebiotics like fructo-oligosaccharides (FOS), found in fruits and vegetables, enhance cognitive function and neuroprotection by modulating the gut microbiota and activating pathways like GLP-1, ameliorating cognitive damage and neurodegeneration in AD models. Preclinical studies show that prebiotic interventions improve cognitive and spatial memory in AD mice, highlighting their therapeutic potential [185].

Synbiotics have emerged as a novel strategy to modulate the gut–brain axis and mitigate AD progression. Preclinical studies show that synbiotic supplementation can improve cognitive function, reduce Aβ accumulation, promote neurogenesis, and decrease inflammation in transgenic AD animal models. These benefits are due to the combined effects of probiotics and prebiotics in regulating the gut microbial composition and signaling pathways involved in AD pathogenesis [186].

FMT aims to improve gut health by introducing donor feces into patients’ gastrointestinal tracts, enhancing microbial diversity and function. Research shows FMT’s potential in treating NDs. In animal models, FMT from aged or AD-affected donors induced cognitive impairment and increased Aβ plaques in younger counterparts, implicating gut microbiota in AD pathology [36]. In humans with cognitive impairment, small studies and case reports suggest FMT can improve cognition. For example, an 82-year-old male’s MMSE score improved from 20 to 29, and a 90-year-old woman also showed cognitive gains after FMT [36,187,188].

## 7. Conclusions

NDs are multifactorial diseases that result from genetic, environmental, and lifestyle factors, with aging as a significant risk factor. Despite varied etiologies, many NDs share common pathological features with abnormal protein aggregation and neuroinflammation as pivotal factors. What is even more intriguing is the active involvement of the peripheral immune system, which can exacerbate neuroinflammation by activating various components, including the complement system, oxidative stress, and pro-inflammatory cytokines, while increasing BBB permeability.

The human intestinal tract contains a huge, active, and complex community of microorganisms. In addition to their central role in regulating the digestive process, commensal bacteria within the gut microbiota, together with their products, have a critical role in regulating the development, homeostasis, and function of innate and adaptive immune cells. Dysbiosis of the gut microbiota has been implicated in numerous disorders, ranging from intestinal diseases, such as colorectal cancer and IBD, to more systemic diseases, such as diabetes, atherosclerosis, and cystic fibrosis.

The MGB axis is the bidirectional pathway of communication involving immune (cytokines, chemokines), hormonal, and neural communication (neuropeptides, neurotransmitters). Given the significant role of neuroinflammation and the immune system in NDs, the gut microbiota’s involvement in NDs can be viewed through its impact on immune system activation.

Healthy gut microbiota and its metabolites like SCFAs and AhR ligands profoundly influence the development and function of innate and adaptive immune cells. In both PD and AD diseases, dysbiosis may increase inflammatory cytokines and bacterial metabolites, which may alter the gut and BBB permeability and cause neuroinflammation. Increased intestinal permeability, also known as “leaky gut”, induces systemic inflammation and production of pro-inflammatory cytokines such as IL-1β, IL-6, and TNF-α that can cross the BBB together with altered immune cells, like monocytes. In CNS, these immune cells behave like polarized M1 cells that, in the chronic phase, reinforce pathological altered microglial cells and astrocytes. However, the two ND pathologies investigated are highly complex, and MGB-immune cell interactions can be differently regulated; indeed, in PD patients, α-syn is the main protein involved in CNS accumulation, but it can also be found in the gastrointestinal tract nerve plexuses, while in AD pathogenesis in the CNS, there is an accumulation of extracellular Aβ peptide and intracellular NFTs composed of hyperphosphorylated tau protein.

Recent advances in microbiota research have revolutionized our understanding of the MGB axis. This growing field has provided compelling evidence of how gut bacteria influence CNS health, offering new avenues for diagnosing, prognosticating, and treating NDs. There is a pressing necessity to provide a better way for diagnosis, prognosis, and therapeutic treatment for NDs. In contrast to other pathologies in NDs, the main pathological burden is confined to the CNS, and very frequently, pathology begins several years before symptoms arise; hence, the diagnosis is made when considerable and irreversible neuronal damage has already occurred. Discoveries in microbiota science may unveil novel biomarkers capable of detecting ND-related changes years before clinical symptoms manifest, facilitating earlier intervention and potentially mitigating irreversible neuronal damage.

Exploring the role of gut microbiota in NDs also underscores the interplay among nutrition, microbiota, and disease. Emerging therapies such as novel antibiotics, prebiotics, probiotics, and microbiota transplants could offer innovative alternatives to traditional treatments. Actively managing the gut microbiota holds promise not only for treating acquired diseases but also for preventive medicine, potentially transforming how we approach and manage NDs in the future.

In conclusion, the evolving understanding of the microbiota–gut–brain axis represents a paradigm shift in neurodegenerative disorder research. The intricate interplay between the gut microbiota, immune system, and CNS highlights promising avenues for early diagnosis, prognostic markers, and novel therapeutic strategies. By harnessing the potential of microbiota-targeted interventions, there is considerable optimism for advancing both treatment and prevention strategies for NDs. Future research efforts should continue to explore and elucidate these complex interactions to realize the full therapeutic potential of gut–brain communication in neurological health.

## Figures and Tables

**Figure 1 ijms-25-12164-f001:**
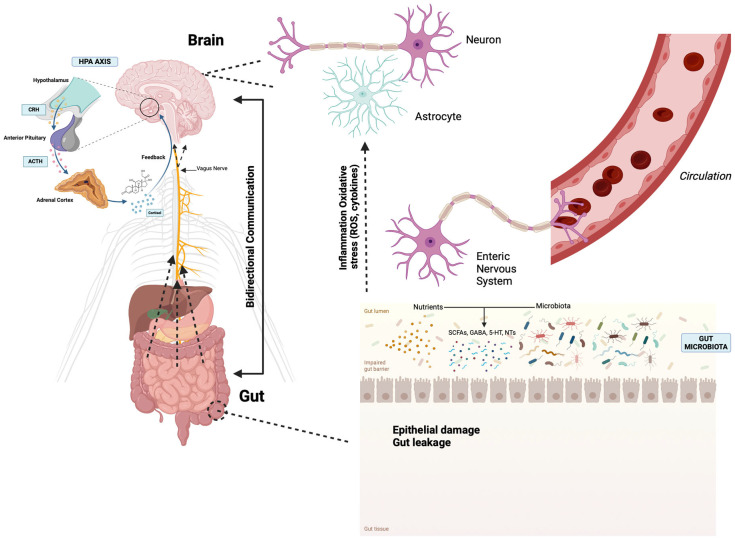
Schematic representation of the microbiota–gut–brain axis, highlighting the complex interactions between the gut microbiota and the brain through neural, immune, and endocrine pathways. The hypothalamic–pituitary–adrenal (HPA) axis is depicted as a key stress-response system, which influences and is influenced by gut microbiota, emphasizing the bidirectional communication between the central nervous system and the gastrointestinal tract. Figure created with Biorender.com (accessed on 1 September 2024).

**Figure 2 ijms-25-12164-f002:**
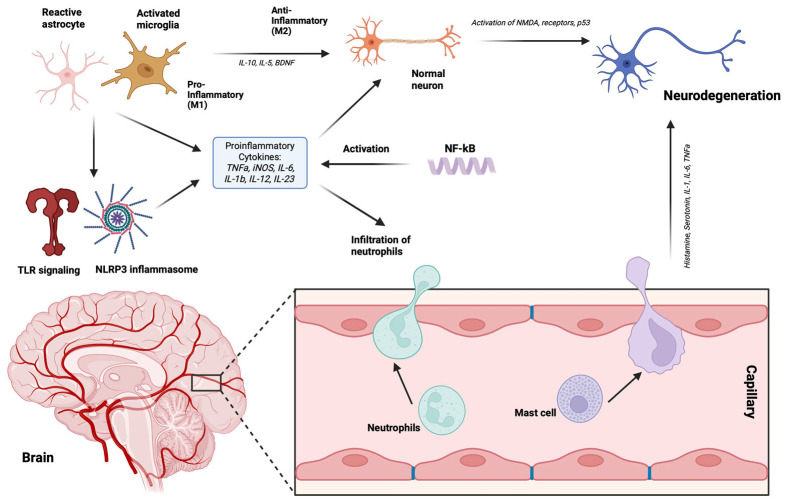
Overview of neuroinflammation highlighting the roles of pro-inflammatory cytokines and microglial polarization. M1 microglia, associated with pro-inflammatory responses, release cytokines that contribute to neurodegeneration, while M2 microglia is involved in anti-inflammatory and tissue repair processes. The balance between these microglial states plays a crucial role in the progression of neurodegenerative diseases. Figure created with Biorender.com (accessed on 1 September 2024).

**Figure 3 ijms-25-12164-f003:**
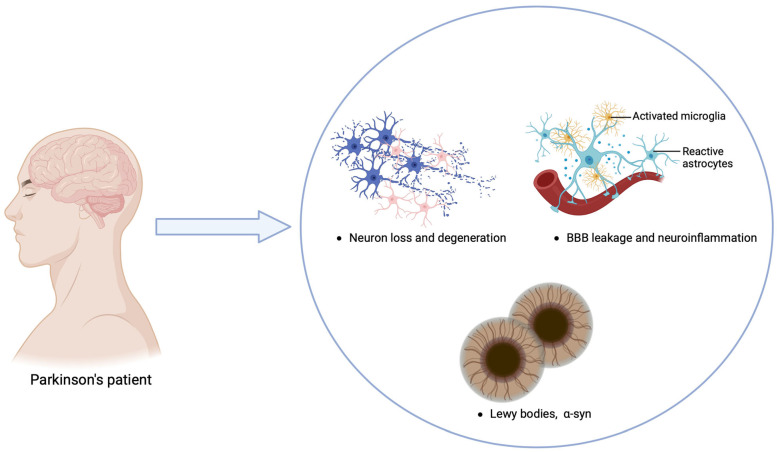
Diagram of the neuropathological features involved in Parkinson’s disease, highlighting key aspects such as BBB leakage, Lewy bodies, neuron loss, and neuroinflammation. Figure created with Biorender.com (accessed on 1 September 2024).

**Figure 4 ijms-25-12164-f004:**
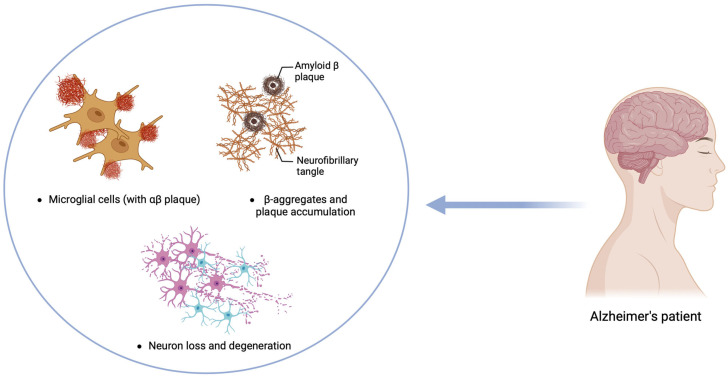
Diagram of the neuropathological features involved in Alzheimer’s disease, highlighting key aspects such as microglial cells with αβ plaque, β aggregates, neuron loss, and degeneration. Figure created with Biorender.com (accessed on 1 September 2024).

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
