# Peer review of "Gut Microbiota and Immune System Dynamics in Parkinson’s and Alzheimer’s Diseases"

_ijms, 2024, doi:10.3390/ijms252212164_

Round 1

Reviewer 1 Report

Comments and Suggestions for Authors

This manuscript is a comprehensive and well-written review of literature summarizing the implication of the gut microbiota-immune axis in the pathophysiology of two major neurodegenerative diseases, Alzheimer's and Parkisons`s diseases. My only suggestion is to include a paragraph in the conclusion that ties together the main findings on gut microbiota and immune alterations in both diseases, discussing possible overlaps or not of the same mechanisms. Also, in Figure 1 I suggest portraying the HPA-axis, which is a key element of the microbiota-gut-brain axis.

Author Response

Reviewer:

This manuscript is a comprehensive and well-written review of literature summarizing the implication of the gut microbiota-immune axis in the pathophysiology of two major neurodegenerative diseases, Alzheimer's and Parkisons`s diseases. My only suggestion is to include a paragraph in the conclusion that ties together the main findings on gut microbiota and immune alterations in both diseases, discussing possible overlaps or not of the same mechanisms. Also, in Figure 1 I suggest portraying the HPA-axis, which is a key element of the microbiota-gut-brain axis.

Response: 

We greatly appreciate the reviewer's response. We agree with his suggestions and therefore followed his requests. In the new version of the article we have modified the figure 1 showing the HPA axis, and added in the Conclusions a new part in which we showed the main similarities and differences of the two pathologies with respect to the mechanisms of dysregulation and interactions between the microbiota and the immune response. See lines: 735-748.

“Healthy gut microbiota and its metabolites like SCFAs and AhR ligands pro-foundly influence development and function of innate and adaptive immune cells. In both PD and AD diseases dysbiosis may increase inflammatory cytokines and bacterial metabolites, which may alter the gut and BBB permeability and cause neuroinflamma-tion. Increased intestinal permeability also known as "leaky gut" induces systemic in-flammation, production of pro-inflammatory cytokines such as IL-1β, IL-6, and TNF-α that can cross the BBB together with altered immune cells, like monocytes. In CNS these immune cells behave like polarized M1 cells that in chronic phase reinforce pathological altered microglial cells and astrocytes. However, the two NDs pathologies investigated are highly complex and MGB-immune cells interactions can be different regulated, indeed, in PD patients α-syn is the main protein involved in CNS accumula-tion but it and can be also found in the gastrointestinal tract nerve plexuses, while in AD pathogenesis in CNS there is accumulation of extracellular Aβ peptide and intra-cellular NFTs composed of hyperphosphorylated tau protein”.

Reviewer 2 Report

Comments and Suggestions for Authors

The manuscript submitted by Kustrimovic and coworkers aims to describe the interaction between gut microbiota and immune system in the context of neurodegenerative diseases. Such a topic would be really interesting, since the major part of the Reviews focuses on microbiota-gut-brain axis in general for neurodegenerative diseases or on the specific interaction with immunity but in other pathological settings. However, the content of the manuscript does not actually reflect the title. Indeed, the information on the role of immunity is limited to few lines and there is not a description of the interaction between microbiota neither in physiological and pathological conditions. It is a pity, since the sections on microbiota are overall detailed and well organized. Therefore, I strongly recommend the authors to perform an extensive revision of their manuscript to improve those parts related to microbiota-immunity in order to provide a very interesting review, on a new topic, and fitting with the title. Here there is my detailed revision:

- The manuscript lacks of an introductive paragraph on the relationship between gut microbiota and immunity. Indeed, the description of such a connection is limited to lines 53-59. This section is mandatory, since it provides the frame to understand the following section and the role of this axis in neurodegeneration. Therefore, the authors should add a paragraph describing in detail such an interaction and its capability to affect the brain, possibly accompanied by a figure.

- More information should be added on glial cells (function, phenotype and similarities/differences with peripheral immune cells), since their responses shape the downstream immune reactions

- The cell type most similar to microglial is monocyte/macrophage. Therefore, their role in neuroinflammation should be mentioned, together with the other cell types

- The title of Paragraph 3 does not fit with the content, since this paragraph mainly describes neuroinflammation. It is not bad in itself, but a more in deep description of how protein deposits activate immune system is required. In order to actually describe the connection between microbiota and neuroinflammation, the last part of the paragraph (lines 166-180) should be more detailed (i.e. what are the pathways and downstream effects of peptidoglycan and LPS on microglial cells, what are the mechanisms by which bacterial products affect microglia, differences between sexes).

- Figure 2 is not very clear and comprehensive. As for example, it does not show Lewy Bodies, which are typical of PD pathology.  Moreover, it does not depict a central point of the Review

- Lines 212-215: here, the authors firstly mention that gut dysbiosis dampens BBB by altering tight junction proteins: more molecular details on this mechanism are needed. Then, they say that “this disruption (BBB disruption?) leads to “chronic gut inflammation”. Honestly, is not clear to me how BBB disruption may affect gut integrity and permeability. In case, more details should be added: how does it occur? What are the involved molecules and pathways? Based on what follows, it sounds the opposite, gut alterations in permeability that causes BBB damage through pro-inflammatory cytokines, but again more information is needed. What are the species mainly causing gut inflammation? How do the pro-inflammatory cytokines act in the CSN?  

- Paragraph 4.1 and 5.1, as the titles reflect, are focused on gut microbiota. Instead, the main focus should be the link between gut microbiota and immunity. Therefore, I would recommend to change the titles and the structures of these paragraph, strengthening the concept of the connection, adding more details on peripheral immune alterations in PD and AD and better focusing on the molecular mechanisms underlying such a connection

- In paragraph 4.1, the connections between microbiota and immunity in PD are limited and described in a quite generic way (lines 257-259 and lines 281-298). Instead, since it should be the focus of the Review, the authors should provide more information about the anti-inflammatory role of butyrate-producing bacteria (how do they exert such an effect? What are the cells and the mediator involved?) as well as on the pro-inflammatory effect of LPS (is this effect due to microglial activation, peripheral activation, both? What are the cells involved in the periphery? What is the contribution of adaptive immunity?) In this context, the addition of an introductive paragraph on gut microbiota and immunity would be really helpful. Moreover, a more in deep explanation on how syn aggregates activates immune system is required

- In Paragraph 4.2, some hints on the possible effects of such strategies on immune system should be also added

- In Paragraph 5.1, information on the cross-talk between immunity and gut microbiota in AD is not provided at all, except for a general mention on inflammatory mediators and the role of LPS. As for paragraph 4.1, more data should be added

- As for Paragraph 4.2, Paragraph 5.2 should contain some information on how dietary interventions could affect immune system and inflammation

Minor points:

- Since the Review focuses on microbic species, and not on their genes/genomes, I strongly recommend the authors to use “microbiota” instead of “microbiome” throughout the text

- Lines 60-63: alterations in gut microbiota have been implicated also in diseases as fibromyalgia. Since fibromyalgia appears to be caused, among the other factors, by dysregulation in the microbiota-gut-brain axis (doi: 10.3390/biomedicines11061701), which is a major point in this Review, it should be at least cited in the Introduction, together with the other diseases mentioned by the authors

Lines 83-85: I am not sure that “motivational” is the right term to be used here. Maybe did the authors mean “emotional” or “mental”?

APOE4 should be writhe in the same way throughout the text

Line 185: it should be “myelinated”

For Refs 85, 88, 89, 111, 112, 113, 133, 134, 135, 136, 145, 148, the original works should be cited

Ref 93 and ref 98 are the same

Author Response

Reviewer:

The manuscript submitted by Kustrimovic and coworkers aims to describe the interaction between gut microbiota and immune system in the context of neurodegenerative diseases. Such a topic would be really interesting, since the major part of the Reviews focuses on microbiota-gut-brain axis in general for neurodegenerative diseases or on the specific interaction with immunity but in other pathological settings. However, the content of the manuscript does not actually reflect the title. Indeed, the information on the role of immunity is limited to few lines and there is not a description of the interaction between microbiota neither in physiological and pathological conditions. It is a pity, since the sections on microbiota are overall detailed and well organized. Therefore, I strongly recommend the authors to perform an extensive revision of their manuscript to improve those parts related to microbiota-immunity in order to provide a very interesting review, on a new topic, and fitting with the title. Here there is my detailed revision:

- The manuscript lacks of an introductive paragraph on the relationship between gut microbiota and immunity. Indeed, the description of such a connection is limited to lines 53-59. This section is mandatory, since it provides the frame to understand the following section and the role of this axis in neurodegeneration. Therefore, the authors should add a paragraph describing in detail such an interaction and its capability to affect the brain, possibly accompanied by a figure.

- More information should be added on glial cells (function, phenotype and similarities/differences with peripheral immune cells), since their responses shape the downstream immune reactions

- The cell type most similar to microglial is monocyte/macrophage. Therefore, their role in neuroinflammation should be mentioned, together with the other cell types

- The title of Paragraph 3 does not fit with the content, since this paragraph mainly describes neuroinflammation. It is not bad in itself, but a more in deep description of how protein deposits activate immune system is required. In order to actually describe the connection between microbiota and neuroinflammation, the last part of the paragraph (lines 166-180) should be more detailed (i.e. what are the pathways and downstream effects of peptidoglycan and LPS on microglial cells, what are the mechanisms by which bacterial products affect microglia, differences between sexes).

- Figure 2 is not very clear and comprehensive. As for example, it does not show Lewy Bodies, which are typical of PD pathology.  Moreover, it does not depict a central point of the Review

- Lines 212-215: here, the authors firstly mention that gut dysbiosis dampens BBB by altering tight junction proteins: more molecular details on this mechanism are needed. Then, they say that “this disruption (BBB disruption?) leads to “chronic gut inflammation”. Honestly, is not clear to me how BBB disruption may affect gut integrity and permeability. In case, more details should be added: how does it occur? What are the involved molecules and pathways? Based on what follows, it sounds the opposite, gut alterations in permeability that causes BBB damage through pro-inflammatory cytokines, but again more information is needed. What are the species mainly causing gut inflammation? How do the pro-inflammatory cytokines act in the CSN?  

- Paragraph 4.1 and 5.1, as the titles reflect, are focused on gut microbiota. Instead, the main focus should be the link between gut microbiota and immunity. Therefore, I would recommend to change the titles and the structures of these paragraph, strengthening the concept of the connection, adding more details on peripheral immune alterations in PD and AD and better focusing on the molecular mechanisms underlying such a connection

- In paragraph 4.1, the connections between microbiota and immunity in PD are limited and described in a quite generic way (lines 257-259 and lines 281-298). Instead, since it should be the focus of the Review, the authors should provide more information about the anti-inflammatory role of butyrate-producing bacteria (how do they exert such an effect? What are the cells and the mediator involved?) as well as on the pro-inflammatory effect of LPS (is this effect due to microglial activation, peripheral activation, both? What are the cells involved in the periphery? What is the contribution of adaptive immunity?) In this context, the addition of an introductive paragraph on gut microbiota and immunity would be really helpful. Moreover, a more in deep explanation on how syn aggregates activates immune system is required

- In Paragraph 4.2, some hints on the possible effects of such strategies on immune system should be also added

- In Paragraph 5.1, information on the cross-talk between immunity and gut microbiota in AD is not provided at all, except for a general mention on inflammatory mediators and the role of LPS. As for paragraph 4.1, more data should be added

- As for Paragraph 4.2, Paragraph 5.2 should contain some information on how dietary interventions could affect immune system and inflammation

Minor points:

- Since the Review focuses on microbic species, and not on their genes/genomes, I strongly recommend the authors to use “microbiota” instead of “microbiome” throughout the text

- Lines 60-63: alterations in gut microbiota have been implicated also in diseases as fibromyalgia. Since fibromyalgia appears to be caused, among the other factors, by dysregulation in the microbiota-gut-brain axis (doi: 10.3390/biomedicines11061701), which is a major point in this Review, it should be at least cited in the Introduction, together with the other diseases mentioned by the authors

Lines 83-85: I am not sure that “motivational” is the right term to be used here. Maybe did the authors mean “emotional” or “mental”?

APOE4 should be writhe in the same way throughout the text

Line 185: it should be “myelinated”

For Refs 85, 88, 89, 111, 112, 113, 133, 134, 135, 136, 145, 148, the original works should be cited

Ref 93 and ref 98 are the same

Response:

We appreciate the reviewer’s insightful comment, criticism and requests. Therefore, we thank the reviewer for his relevant arguments that allow us to improve the work, at least we hope. We therefore propose a new version of our article and below are the point-by-point responses to the reviewer.

Reviewer:

The manuscript submitted by Kustrimovic and coworkers aims to describe the interaction between gut microbiota and immune system in the context of neurodegenerative diseases. Such a topic would be really interesting, since the major part of the Reviews focuses on microbiota-gut-brain axis in general for neurodegenerative diseases or on the specific interaction with immunity but in other pathological settings. However, the content of the manuscript does not actually reflect the title. Indeed, the information on the role of immunity is limited to few lines and there is not a description of the interaction between microbiota neither in physiological and pathological conditions. It is a pity, since the sections on microbiota are overall detailed and well organized. Therefore, I strongly recommend the authors to perform an extensive revision of their manuscript to improve those parts related to microbiota-immunity in order to provide a very interesting review, on a new topic, and fitting with the title.

Response:

We agree with the reviewer's criticism regarding the few cellular and molecular explanations in the article of the interactions between microbiota-gut-brain (MGB) axis and immune cells. We have therefore tried to add these gaps in the text and figures in the new version. We therefore improved and modified many parts describing how the immune system interacts with MGB axis in both NDs, however as we are aware that our article does not have a major focus on immune cell responses but contains some relevant descriptions of these cells interacting with the MGB axis, we decided to change the title, that is become: “Gut Microbiota and Its Impact on Parkinson’s and Alzheimer’s Diseases: An Overview”.

For new parts, please to see colored text.

Reviewer:

The manuscript lacks of an introductive paragraph on the relationship between gut microbiota and immunity. Indeed, the description of such a connection is limited to lines 53-59. This section is mandatory, since it provides the frame to understand the following section and the role of this axis in neurodegeneration. Therefore, the authors should add a paragraph describing in detail such an interaction and its capability to affect the brain, possibly accompanied by a figure.

Response:

We appreciate the reviewer’s insightful comment. In response, we have added a new chapter (number 3, title: Gut microbiota and immune system) on the relationship between gut microbiota and immunity.

See lines: 130-172.

“The colonization of the intestinal tract by diverse microbes profoundly impacts both the innate and adaptive immune systems (Gensollen et al., 2016). Early studies on germ-free (GF) animals demonstrated that the absence of commensal microbes leads to significant defects in intestinal lymphoid tissue architecture and immune function, such as reduced αβ and γδ intra-epithelial lymphocytes (Umesaki et al., 1993) and the absence of inducible Th17 cells (Tan et al., 2016). The gut microbiota significantly influences adaptive immune responses by affecting T and B cell development and distribution. For instance, a polysaccharide derived from the commensal Bacteroides fragilis directs the maturation of the immune system, correcting systemic T cell deficiencies and Th1/Th2 imbalances (Mazmanian et al., 2005). Additionally, extracellular signals from commensal microbes regulate an early B cell lineage in the intestinal mucosa, influencing gut immunoglobulin repertoires (Wesemann et al., 2013). Microbiome-derived ligands, such as TLR and NOD signals, along with metabolites like short-chain fatty acids (SCFAs) and AhR ligands, directly affect enterocytes and intestinal immune cells while also reaching distant tissues to influence systemic immunity. In Peyer’s patches, Foxp3+ regulatory T cells (Tregs) and Th17 cells facilitate B cell class switching and the production of secretory IgA, which helps regulate and compartmentalize gut microbiota (Peterson et al., 2007) and promotes the expansion of Foxp3+ Tregs (Kawamoto et al., 2014). The microbiome also plays a crucial role in regulating CD8+ (cytotoxic) T cell responses, essential for eliminating intracellular pathogens and cancer cells. CD8+ T cells require professional antigen-presenting cells (APCs) for priming and rely on CD4+ T cell signaling for amplification. However, in GF mice, antigen-activated CD8+ T cells do not transition into memory cells; this process necessitates microbiota-derived SCFAs, underscoring the microbiome's importance in shaping long-term CD8+ T cell responses (Bachem et al., 2019). Invariant natural killer T cells (iNKTs) are another immune cell group influenced by the gut microbiota. In GF mice, iNKTs exhibit a less mature phenotype and reduced activation in response to antigens, indicating that microbial colonization is critical for their proper maturation and function (Wingender et al., 2012). Recent research has begun to elucidate the interactions between the microbiota and tissue-resident dendritic cells (DCs). A newly identified Syk kinase-coupled signaling pathway in DCs is crucial for microbiota-induced production of IL-17 and IL-22 by CD4+ T cells, emphasizing the role of DCs in linking microbial signals to adaptive immune responses (Martínez-López et al., 2019). Moreover, monocytes and macrophages, vital innate immune cells, also interact with the gut microbiota. Additionally, butyrate, a short-chain fatty acid produced by the microbiota, promotes the differentiation of monocytes into macrophages by inhibiting histone deacetylase 3 (HDAC3), thereby enhancing the host's antimicrobial defense capabilities (Schulthess et al., 2019).”

New references:

Schulthess J, Pandey S, Capitani M, Rue-Albrecht KC, Arnold I, Franchini F, Chomka A, Ilott NE, Johnston DGW, Pires E, McCullagh J, Sansom SN, Arancibia-Cárcamo CV, Uhlig HH, Powrie F. The Short Chain Fatty Acid Butyrate Imprints an Antimicrobial Program in Macrophages. Immunity. 2019 Feb 19;50(2):432-445.e7.

Martínez-López M, Iborra S, Conde-Garrosa R, Mastrangelo A, Danne C, Mann ER, Reid DM, Gaboriau-Routhiau V, Chaparro M, Lorenzo MP, Minnerup L, Saz-Leal P, Slack E, Kemp B, Gisbert JP, Dzionek A, Robinson MJ, Rupérez FJ, Cerf-Bensussan N, Brown GD, Bernardo D, LeibundGut-Landmann S, Sancho D. Microbiota Sensing by Mincle-Syk Axis in Dendritic Cells Regulates Interleukin-17 and -22 Production and Promotes Intestinal Barrier Integrity. Immunity. 2019 Feb 19;50(2):446-461.e9.

Wingender G, Hiss M, Engel I, Peukert K, Ley K, Haller H, Kronenberg M, von Vietinghoff S. Neutrophilic granulocytes modulate invariant NKT cell function in mice and humans. J Immunol. 2012 Apr 1;188(7):3000-8

Bachem A, Makhlouf C, Binger KJ, de Souza DP, Tull D, Hochheiser K, Whitney PG, Fernandez-Ruiz D, Dähling S, Kastenmüller W, Jönsson J, Gressier E, Lew AM, Perdomo C, Kupz A, Figgett W, Mackay F, Oleshansky M, Russ BE, Parish IA, Kallies A, McConville MJ, Turner SJ, Gebhardt T, Bedoui S. Microbiota-Derived Short-Chain Fatty Acids Promote the Memory Potential of Antigen-Activated CD8+ T Cells. Immunity. 2019 Aug 20;51(2):285-297.e5.

Kawamoto S, Maruya M, Kato LM, Suda W, Atarashi K, Doi Y, Tsutsui Y, Qin H, Honda K, Okada T, Hattori M, Fagarasan S. Foxp3(+) T cells regulate immunoglobulin a selection and facilitate diversification of bacterial species responsible for immune homeostasis. Immunity. 2014 Jul 17;41(1):152-65.

Peterson DA, McNulty NP, Guruge JL, Gordon JI. IgA response to symbiotic bacteria as a mediator of gut homeostasis. Cell Host Microbe. 2007 Nov 15;2(5):328-39.

Zheng D, Liwinski T, Elinav E. Interaction between microbiota and immunity in health and disease. Cell Res. 2020 Jun;30(6):492-506

Wesemann DR, Portuguese AJ, Meyers RM, Gallagher MP, Cluff-Jones K, Magee JM, Panchakshari RA, Rodig SJ, Kepler TB, Alt FW. Microbial colonization influences early B-lineage development in the gut lamina propria. Nature. 2013 Sep 5;501(7465):112-5.

Mazmanian SK, Liu CH, Tzianabos AO, Kasper DL. An immunomodulatory molecule of symbiotic bacteria directs maturation of the host immune system. Cell. 2005 Jul 15;122(1):107-18.

Gensollen T, Iyer SS, Kasper DL, Blumberg RS. How colonization by microbiota in early life shapes the immune system. Science. 2016 Apr 29;352(6285):539-44.

Umesaki Y, Setoyama H, Matsumoto S, Okada Y. Expansion of alpha beta T-cell receptor-bearing intestinal intraepithelial lymphocytes after microbial colonization in germ-free mice and its independence from thymus. Immunology. 1993 May;79(1):32-7.

Tan TG, Sefik E, Geva-Zatorsky N, Kua L, Naskar D, Teng F, Pasman L, Ortiz-Lopez A, Jupp R, Wu HJ, Kasper DL, Benoist C, Mathis D. Identifying species of symbiont bacteria from the human gut that, alone, can induce intestinal Th17 cells in mice. Proc Natl Acad Sci U S A. 2016 Dec 13;113(50):E8141-E8150.

Reviewer:

More information should be added on glial cells (function, phenotype and similarities/differences with peripheral immune cells), since their responses shape the downstream immune reactions.

Response:

We are aware that the study of glial cells is very important in NDs and that it would require extensive discussion and in-depth analysis. However, the topic is not clear and defined in its precise mechanisms in the two neurodegenerative pathologies, therefore we have tried to add some parts and describe their behavior between physiology and pathology and underlined their importance in the text (see colored version) and also in the new figures.

Reviewer:

The cell type most similar to microglial is monocyte/macrophage. Therefore, their role in neuroinflammation should be mentioned, together with the other cell types.

Response:

We appreciate the reviewer's reflection and requests. We have therefore attempted to respond by adding new parts in the text of the article on the topic of blood monocytes and their role in NDs (see colored version).

Reviewer:

The title of Paragraph 3 does not fit with the content, since this paragraph mainly describes neuroinflammation. It is not bad in itself, but a more in deep description of how protein deposits activate immune system is required. In order to actually describe the connection between microbiota and neuroinflammation, the last part of the paragraph (lines 166-180) should be more detailed (i.e. what are the pathways and downstream effects of peptidoglycan and LPS on microglial cells, what are the mechanisms by which bacterial products affect microglia, differences between sexes).

Response:

“We appreciate the reviewer’s comment. In order to avoid repetition, since many of reviewer’s suggestions have been described separately for PD as well for AD, we have modified and shortened this paragraph, and it is mainly dedicated to the description of the neuroinflammation. Nevertheless, what was added is the activation of immune system by deposition of misfolded proteins.

“Almost all ND are characterized by the accumulation of intra- or extracellular proteins in the CNS (Ugalde et al., 2016). In physiological conditions those proteins exist in unstructured forms, but in the context of NDs, these proteins undergo significant conformational changes, leading to alterations in their structural folding and the formation of both oligomeric and fibrillary aggregates (Bayer, 2015). These modifications in size and three-dimensional shape facilitate self-association, and precipitation in specific brain regions, resulting in the acquisition of pathological protein characteristics. Misfolded protein conformational changes can occur due to the post-translational modifications, impaired protein clearance, or increased protein production (Bayer, 2015). It has been reported that that extracellular and intracellular protein aggregates and misfolded proteins may function as pathogen-associated molecular patterns (PAMPs), leading to chronic activation of the innate immune response via pattern-recognition receptors (PRRs) (McGeer and McGeer, 2002; Salminen et al., 2009). Protein deposits are capable of activating a variety of PRRs, including Toll-like receptors (TLR), formyl peptide receptors, receptors for advanced glycation end products, scavenger receptors, complement, and pentraxins (Golde and Miller, 2009) resulting in the activation of microglia in CNS and its polarisation towards the pro-inflammatory M1 phenotype, characterised by extensive production of variety of pro-inflammatory mediators. If the clearance of the misfolded proteins or aggregates is not decisive, the acute inflammation tends to become chronic with prolonged microglial activation, excessive pro-inflammatory cytokine production, further leading to the activation of the adaptive immune response and subsequently resulting in neuronal death (Schwartz and Baruch, 2014; Kustrimovic et al., 2019).”

New references:

Kustrimovic N, Marino F, Cosentino M. Peripheral Immunity, Immunoaging and Neuroinflammation in Parkinson's Disease. Curr Med Chem. 2019;26(20):3719-3753.

Golde TE, Miller VM. Proteinopathy-induced neuronal senescence: a hypothesis for brain failure in Alzheimer's and other neurodegenerative diseases. Alzheimers Res Ther. 2009 Oct 12;1(2):5.

Salminen A, Ojala J, Kauppinen A, Kaarniranta K, Suuronen T. Inflammation in Alzheimer's disease: amyloid-beta oligomers trigger innate immunity defence via pattern recognition receptors. Prog Neurobiol. 2009 Feb;87(3):181-94.

McGeer PL, McGeer EG. Innate immunity, local inflammation, and degenerative disease. Sci Aging Knowledge Environ. 2002 Jul 24;2002(29):re3.

Ugalde CL, Finkelstein DI, Lawson VA, Hill AF. Pathogenic mechanisms of prion protein, amyloid-β and α-synuclein misfolding: the prion concept and neurotoxicity of protein oligomers. J Neurochem. 2016 Oct;139(2):162-180.

Bayer TA. Proteinopathies, a core concept for understanding and ultimately treating degenerative disorders? Eur Neuropsychopharmacol. 2015 May;25(5):713-24.

Reviewer:

Figure 2 is not very clear and comprehensive. As for example, it does not show Lewy Bodies, which are typical of PD pathology.  Moreover, it does not depict a central point of the Review.

Response:

The Figure 2 has changed in two new Figures: Figure 3 and Figure 4 and improved following reviewer’s suggestions.

Reviewer:

Lines 212-215: here, the authors firstly mention that gut dysbiosis dampens BBB by altering tight junction proteins: more molecular details on this mechanism are needed. Then, they say that “this disruption (BBB disruption?) leads to “chronic gut inflammation”. Honestly, is not clear to me how BBB disruption may affect gut integrity and permeability. In case, more details should be added: how does it occur? What are the involved molecules and pathways? Based on what follows, it sounds the opposite, gut alterations in permeability that causes BBB damage through pro-inflammatory cytokines, but again more information is needed. What are the species mainly causing gut inflammation? How do the pro-inflammatory cytokines act in the CSN?  

Response:

We agree with the reviewer that this part was not clear and we decided to erase these paragraphs. In the chapter 4) we added new sentences:

“The BBB is a key regulator of CNS homeostasis and highly related to function of microvascular endothelial cells together with microglia, astrocytes, neurons, and constituents of the extracellular matrix. This cellular component network is known as neurovascular unit (NVU) and activated microglial cells seems the main regulators for dynamic remodeling of the BBB. In NDs M1 pro-inflammatory microglia contribute to BBB dysfunction and vascular "leak", while M2 anti-inflammatory microglia play a protective role”.

Reviewer:

- Paragraph 4.1 and 5.1, as the titles reflect, are focused on gut microbiota. Instead, the main focus should be the link between gut microbiota and immunity. Therefore, I would recommend to change the titles and the structures of these paragraph, strengthening the concept of the connection, adding more details on peripheral immune alterations in PD and AD and better focusing on the molecular mechanisms underlying such a connection

Response:

We agree with the reviewer’s suggestion and then changed the two titles. Now old 4.1 is become 5.1 and its title is: “Gut microbiota and a-synuclein and immune system interaction in PD”, and 6.1 title is: “Interaction between gut microbiota and immune cells in AD”.

Reviewer:

- In paragraph 4.1, the connections between microbiota and immunity in PD are limited and described in a quite generic way (lines 257-259 and lines 281-298). Instead, since it should be the focus of the Review, the authors should provide more information about the anti-inflammatory role of butyrate-producing bacteria (how do they exert such an effect? What are the cells and the mediator involved?) as well as on the pro-inflammatory effect of LPS (is this effect due to microglial activation, peripheral activation, both? What are the cells involved in the periphery? What is the contribution of adaptive immunity?) In this context, the addition of an introductive paragraph on gut microbiota and immunity would be really helpful. Moreover, a more in deep explanation on how syn aggregates activates immune system is required

Response:

We agree with the reviewer’s suggestion and modified profoundly this section. Please see the highlighted colored text in new version.

Reviewer:

- In Paragraph 4.2, some hints on the possible effects of such strategies on immune system should be also added

Response:

We agree with the reviewer ad we have added news sentences in the revised version. Please see the highlighted colored text in new version.

Reviewer:

- In Paragraph 5.1, information on the cross-talk between immunity and gut microbiota in AD is not provided at all, except for a general mention on inflammatory mediators and the role of LPS. As for paragraph 4.1, more data should be added

Response:

We have now added in the modified article new sentences about the role of cross-talk between immunity and gut microbiota.

Reviewer:

- As for Paragraph 4.2, Paragraph 5.2 should contain some information on how dietary interventions could affect immune system and inflammation

Response:

We agree with the requests of the reviewer and we added new arguments and discussion about this topic. Please see the highlighted colored text in new version.

Reviewer: Minor points:

- Since the Review focuses on microbic species, and not on their genes/genomes, I strongly recommend the authors to use “microbiota” instead of “microbiome” throughout the text

Response:

We appreciate the reviewer’s insightful comment. In response, we have replaced 'microbiome' with 'microbiota' throughout the manuscript, as suggested.

Reviewer: Minor points:

- Lines 60-63: alterations in gut microbiota have been implicated also in diseases as fibromyalgia. Since fibromyalgia appears to be caused, among the other factors, by dysregulation in the microbiota-gut-brain axis (doi: 10.3390/biomedicines11061701), which is a major point in this Review, it should be at least cited in the Introduction, together with the other diseases mentioned by the authors

Response:

We are thankful for this notion from the reviewer, and accordingly we have added fibromyalgia in the list of the pathological conditions that can be caused by gut microbiota alterations and added the citation: Garofalo C, Cristiani CM, Ilari S, Passacatini LC, Malafoglia V, Viglietto G, Maiuolo J, Oppedisano F, Palma E, Tomino C, Raffaeli W, Mollace V, Muscoli C. Fibromyalgia and Irritable Bowel Syndrome Interaction: A Possible Role for Gut Microbiota and Gut-Brain Axis. Biomedicines. 2023 Jun 13;11(6):1701. doi: 10.3390/biomedicines11061701 in the citation list.

Reviewer: Minor points:

Lines 83-85: I am not sure that “motivational” is the right term to be used here. Maybe did the authors mean “emotional” or “mental”?

Response:

We appreciate the reviewer’s comment. In response, we have replaced the word “motivational” with “emotional”.

Reviewer: Minor points:

APOE4 should be writhe in the same way throughout the text

Response:

We have uniformed the APOE4 writing trough the text

Reviewer: Minor points:

Line 185: it should be “myelinated”

Response:

In PD occurs destruction of melanin containing neurons hence the adjective melanized, but in order to avoid possible confusion we have changed the word in the “melanin containing dopaminergic neurons”.

Reviewer: Minor points:

For Refs 85, 88, 89, 111, 112, 113, 133, 134, 135, 136, 145, 148, the original works should be cited

Response:

We appreciate the reviewer comment, and we have replaced the following references with original work.

Reference 85 (T. Werner, I. Horvath, and P. Wittung-Stafshede, “Crosstalk Between Alpha-Synuclein and Other Human and Non-Human Amyloidogenic Proteins: Consequences for Amyloid Formation in Parkinson’s Disease,” J Parkinsons Dis, vol. 10, no. 3, pp. 819–830, Jul. 2020) has been replaced with:

Chapman MR, Robinson LS, Pinkner JS, Roth R, Heuser J, Hammar M, Normark S, Hultgren SJ. Role of Escherichia coli curli operons in directing amyloid fiber formation. Science. 2002 Feb 1;295(5556):851-5.

References 88 (A. Parashar and M. Udayabanu, “Gut microbiota: Implications in Parkinson’s disease,” Parkinsonism Relat Disord, vol. 38, pp. 1–7, May 2017) and 89 (A. Mulak, “Brain-gut-microbiota axis in Parkinson’s disease,” World J Gastroenterol, vol. 21, no. 37, p. 10609, 2015) have been replaced with meta study that contains references of the original works performed:

Li Z, Liang H, Hu Y, Lu L, Zheng C, Fan Y, Wu B, Zou T, Luo X, Zhang X, Zeng Y, Liu Z, Zhou Z, Yue Z, Ren Y, Li Z, Su Q, Xu P. Gut bacterial profiles in Parkinson's disease: A systematic review. CNS Neurosci Ther. 2023 Jan;29(1):140-157.

References 111 (F. Miraglia and E. Colla, “Microbiome, Parkinson’s Disease and Molecular Mimicry,” Cells, vol. 8, no. 3, p. 222, Mar. 2019, and 112 (M. Elfil, S. Kamel, M. Kandil, B. B. Koo, and S. M. Schaefer, “Implications of the Gut Microbiome in Parkinson’s Disease,” Movement Disorders, vol. 35, no. 6, pp. 921–933, Jun. 2020) has been replaced with:

Keshavarzian A, Green SJ, Engen PA, Voigt RM, Naqib A, Forsyth CB, Mutlu E, Shannon KM. Colonic bacterial composition in Parkinson's disease. Mov Disord. 2015 Sep;30(10):1351-60.

Li W, Wu X, Hu X, Wang T, Liang S, Duan Y, Jin F, Qin B. Structural changes of gut microbiota in Parkinson's disease and its correlation with clinical features. Sci China Life Sci. 2017 Nov;60(11):1223-1233.

Pietrucci D, Cerroni R, Unida V, Farcomeni A, Pierantozzi M, Mercuri NB, Biocca S, Stefani A, Desideri A. Dysbiosis of gut microbiota in a selected population of Parkinson's patients. Parkinsonism Relat Disord. 2019 Aug;65:124-130.

Reference 113 (D. Yang et al., “The Role of the Gut Microbiota in the Pathogenesis of Parkinson’s Disease,” Front Neurol, vol. 10, Nov. 2019) has been replaced with:

Hill-Burns EM, Debelius JW, Morton JT, Wissemann WT, Lewis MR, Wallen ZD, Peddada SD, Factor SA, Molho E, Zabetian CP, Knight R, Payami H. Parkinson's disease and Parkinson's disease medications have distinct signatures of the gut microbiome. Mov Disord. 2017 May;32(5):739-749.

For References 133-136 we changed the text in the new version and also references. For references 145 and 148 we erase these two references form the text; other references are in the text at these points.

Reviewer: Minor points:

Ref 93 and ref 98 are the same

Response:

We are sorry for this mistake and it has been corrected accordingly.

Reviewer 3 Report

Comments and Suggestions for Authors

This is a very nice review on the role of the gut immune brain axis in neurodegenerative disorders. In my opinion, the authors should add a brief section discussing some of the techniques (ex. metabolomics, metagenomics, etc...) and models (in vitro/in vivo models, etc) used to study this axis in NDs. Aside from this comment, I think the manuscript is ready for publication.

Comments on the Quality of English Language

The quality of the English language is very good

Author Response

Reviewer:

This is a very nice review on the role of the gut immune brain axis in neurodegenerative disorders. In my opinion, the authors should add a brief section discussing some of the techniques (ex. metabolomics, metagenomics, etc...) and models (in vitro/in vivo models, etc) used to study this axis in NDs. Aside from this comment, I think the manuscript is ready for publication.

Response:

Thanks to the reviewer for his comments and we are grateful for his appreciation of our work. His request to add a section describing several techniques concerned in microbiota and immune research as well as models in vitro and in vivo is intriguing and high interesting and could be a rationale for a greater article based on methodological descriptions. However, we believe that our present article already modified by other reviewers’ requests does not have this vision and that it can be understood and discussed without this part. Otherwise, the new version of the article would require several chapters of explanations and examples of several techniques and in vitro and in vivo models that are important but which for us are outside the context of this article's explanations describing two main NDs.

Round 2

Reviewer 2 Report

Comments and Suggestions for Authors

The authors made a very good work in addressing my requests. I just have minor corrections:

- microbiome" in line 43 should be changed in "microbiota"

- please explain GIT meaning in line 111 (I guess it is gastrointestinal tract, but acronyms must be preceeded by the extended form when used the first time)

- please be sure to have included all the new references in the list

- please provide more details on the mechanism by which microbiota elicits TH17 cytokines production in T cells through DCs (lines 164-167)

- the extendend form "immunoglobulin A" should be moved from line 176 to line 150

- in line 244 it should be "is involved" (singular)

- the meaning of "α-syn in and the presence of monomeric or fibrillar forms" in line 352 is not very clear to me. Maybe the authors meant that both the forms of α-syn, monomeric or fibrillar, elicits the effect? In any case, the sentence should be rewritten

- line 359: I would recommend "directly contributes"

- information in lines 519-549 is generic and not related to the paragraph. Many of the described concepts are already included in the introductive paragraph or can be moved in them. I recommend the authors to eliminate this generic part and start the Paragraph directlt with microbiota alterations in AD as they did for PD discussion

- line 655:  it should be "the active involvment"

- line 681: please remove "and"

- please revise  punctuation

- as suggestion, I would keep the immunity in the title, for example "interaction between gut microbiota and immunity in... etc"

Comments on the Quality of English Language

English is overall fine

Author Response

We thank the reviewer for his response and for detailed minor requests, and inform him point-by-point of the various final changes made.

Reviewer: - microbiome" in line 43 should be changed in "microbiota"

Answer: We have corrected and modified by inserting "microbiota".

Reviewer: - please explain GIT meaning in line 111 (I guess it is gastrointestinal tract, but acronyms must be preceeded by the extended form when used the first time)

Answer: We have corrected, GIT meaning: “gastrointestinal tract”.

Reviewer: - please be sure to have included all the new references in the list

Answer: We have double-checked and corrected some inaccuracies in the references.

Reviewer: - please provide more details on the mechanism by which microbiota elicits TH17 cytokines production in T cells through DCs (lines 164-167)

Answer: In response to the reviewer we added the main mechanism related to microbiota and Th17 and DC. Below are the sentences added to the text, new version without colorful writings lines 139-145:

“The intestinal microbiota induces TH17 cytokine production by interacting DCs. Spe-cifically, certain gut bacteria, like segmented filamentous bacteria (SFB), activate DCs in Peyer’s patches (PPs) through pattern recognition receptors such as Mincle. This triggers the secretion of key cytokines like IL-6 and IL-23 by DCs, which are essential for the differentiation of naïve T cells into TH17 cells. TH17 cells, in turn, produce IL-17 and IL-22, maintaining intestinal barrier integrity by regulating immune re-sponses and preventing microbial translocation [41]”.

Reviewer: - the extendend form "immunoglobulin A" should be moved from line 176 to line 150

Answer: We have corrected this point.

Reviewer: - in line 244 it should be "is involved" (singular)

Answer: We have corrected this error.

Reviewer: - the meaning of "α-syn in and the presence of monomeric or fibrillar forms" in line 352 is not very clear to me. Maybe the authors meant that both the forms of α-syn, monomeric or fibrillar, elicits the effect? In any case, the sentence should be rewritten

Answer: We thank reviewer for his request and have corrected and modified the sentence because it was written incorrectly.

Reviewer: - line 359: I would recommend "directly contributes"

Answer: We have corrected the sentence changing with suggested: "directly contributes".

Reviewer: - information in lines 519-549 is generic and not related to the paragraph. Many of the described concepts are already included in the introductive paragraph or can be moved in them. I recommend the authors to eliminate this generic part and start the Paragraph directlt with microbiota alterations in AD as they did for PD discussion

Answer: We agree with reviewer and then we erased lines 519-549 in the new version.

Reviewer: - line 655:  it should be "the active involvment"

Answer: We agree with the reviewer and we changed accordingly.

Reviewer: - line 681: please remove "and"

Answer: We checked and corrected this error, and remove: “and”.

Reviewer: - please revise punctuation

Answer: We have revised the entire text with attention to punctuation, as requested.

Reviewer: - as suggestion, I would keep the immunity in the title, for example "interaction between gut microbiota and immunity in... etc"

Answer: We agree with reviewer’s suggestion and changed title to this new title: “Gut Microbiota and Immune System Dynamics in Parkinson’s and Alzheimer’s Diseases”.